# The Role of Anode Potential in Electromicrobiology

**DOI:** 10.3390/microorganisms13030631

**Published:** 2025-03-11

**Authors:** Yanran Li, Yiwu Zong, Chunying Feng, Kun Zhao

**Affiliations:** 1School of Chemical Engineering and Technology, Tianjin University, Tianjin 300072, China; liyanran@tju.edu.cn; 2State Key Laboratory of Synthetic Biology, and Frontiers Science Center for Synthetic Biology, Tianjin 300000, China; 3Frontiers Research Institute for Synthetic Biology, Tianjin University, Tianjin 301799, China; 4Jiangsu Key Laboratory of Sericultural and Animal Biotechnology, School of Biotechnology, Jiangsu University of Science and Technology, Zhenjiang 212100, China; fcy0315@tju.edu.cn; 5Key Laboratory of Silkworm and Mulberry Genetic Improvement, Ministry of Agriculture and Rural Affairs, Sericultural Scientific Research Center, Chinese Academy of Agricultural Sciences, Zhenjiang 212100, China; 6The Sichuan Provincial Key Laboratory for Human Disease Gene Study, and The Institute of Laboratory Medicine, Sichuan Provincial People’s Hospital, University of Electronic Science and Technology of China, Chengdu 610054, China; 7Institute of Fundamental and Frontier Sciences, University of Electronic Science and Technology of China, Chengdu 610054, China

**Keywords:** anode potential, electroactive microorganisms, bioelectricity generation, microbial physiology

## Abstract

Electroactive microorganisms are capable of exchanging electrons with electrodes and thus have potential applications in many fields, including bioenergy production, microbial electrochemical synthesis of chemicals, environmental protection, and microbial electrochemical sensors. Due to the limitations of low electron transfer efficiency and poor stability, the application of electroactive microorganisms in industry is still confronted with significant challenges. In recent years, many studies have demonstrated that modulating anode potential is one of the effective strategies to enhance electron transfer efficiency. In this review, we have summarized approximately 100 relevant studies sourced from PubMed and Web of Science over the past two decades. We present the classification of electroactive microorganisms and their electron transfer mechanisms and elucidate the impact of anode potential on the bioelectricity behavior and physiology of electroactive microorganisms. Our review provides a scientific basis for researchers, especially those who are new to this field, to choose suitable anode potential conditions for practical applications to optimize the electron transfer efficiency of electroactive microorganisms, thus contributing to the application of electroactive microorganisms in industry.

## 1. Introduction

Electromicrobiology is an emerging sub-discipline of microbiology that explores the interactions between microorganisms and electronic conductors, as well as the novel electrical properties of microorganisms [1,2]. These microorganisms are commonly referred to as electroactive microorganisms. Electroactive microorganisms are capable of exchanging electrons with electrodes to generate electric current and then consume electric current to power their respiration [2]. In recent years, with the exploration, selection, mechanistic elucidation, and function enhancement of electroactive microorganisms, their potential applications in bioenergy production (such as electric energy [3,4] and hydrogen energy [5,6]), microbial electrochemical synthesis (such as biofuels [7], bioplastics [8] and inorganic nanomaterials [9]), bioremediation (degradation of pollutants [10,11]), biosensors (environmental monitoring and toxicity detection [12,13]) and other fields have gradually become clear, holding profound significance for sustainable technology and green chemistry. However, the industrial application of electroactive microorganisms still faces a number of challenges. Firstly, the output power of current systems such as microbial fuel cells (MFCs) is relatively low, which limits their application in large-scale power generation. Secondly, the high internal resistance of the system leads to reduced energy transfer efficiency and upraised energy loss. In addition, the start-up time of the system is long, which affects the efficiency of the continuous production processes. The stability and environmental adaptability of the microbial community are also key issues, as the fluctuation of the microbial community may affect the performance and stability of the system. Addressing these challenges requires a deeper understanding of the diversity and complexity of the metabolism of electroactive microorganisms and the mechanisms of their interaction with the environment, especially with the electrodes. Such insights are essential to innovate the technologies that foster the development of microbial electrochemical technology, providing more exploration and technical support for the design and construction of novel microbial electrochemical devices.

The dynamic interactions between microbial cells and electrodes can be characterized as either capacitive or Faradaic in nature (Figure 1) [1]. Capacitive interactions involve changes in the thickness of the double layer composed of ions and water as the distance between the electrode and the cell varies during the attachment and detachment processes, resulting in capacitance changes, while Faradaic interactions involve the redox reactions of microbial cells and of any molecular species that participate in extracellular electron exchange processes [1]. In microbial electrochemistry, the former is significant in the enrichment of electroactive microorganisms on the electrode surface, which affects the formation of biofilms and the start-up time of microbial electrochemical systems (MESs). The latter is crucial for microbial metabolism and electron transfer to electrodes. Undoubtedly, the appropriately applied potential will play an important role in enhancing these two processes.

With the development of microbial electrochemical technology in the fields of sustainable energy production and environmental remediation, more and more studies have demonstrated the critical role of setting anodic potential, which can regulate microbial metabolic pathways, enhance electron transfer efficiency, and improve the overall performance of the system [14,15]. However, it is important to note that different MESs may require different optimal anode potential settings, which depend on the types of electroactive microorganisms, metabolic activities, and even details of the system. Therefore, to achieve the optimal performance of MESs, precise control and optimization of the anodic potential are essential. This includes not only the adjustment of potential values but also the refinement of potential control strategies and a deeper understanding of the mechanisms underlying the effects of electric potential on microorganisms. Through these endeavors, we may inspire the potential of electroactive microorganisms and facilitate the application of microbial electrochemical technology in areas such as energy production, environmental remediation, and biosynthesis.

In this review, we summarize the characteristics of electroactive microorganisms and delve into the significant effects of anode potential on the physiology and bioelectricity of electroactive microorganisms, including electron transfer pathways of electroactive microorganisms and the selection of electroactive microbial communities in bioelectricity, and mobility, adhesion, growth, metabolic pathways and biofilms of electroactive microorganisms in physiology. Our review provides a scientific basis for researchers to choose suitable anode potential conditions in practical applications to optimize the electron transfer efficiency of electroactive microorganisms, thus contributing to the application of electroactive microorganisms in industry.

## 2. Classification and Characteristics of Electroactive Microorganisms

Electroactive microorganisms discovered so far cover all three domains of life (Figure 2)—bacteria (accounting for the vast majority of the known electroactive microorganisms, including the Firmicutes like *Lactococcus lactis*, the Actinobacteria phylum like *Micrococcus luteus*, and all classes of Proteobacteria like *Geobacter sulfurreducens* and *Shewanella oneidensis*); archaea (such as methanogens, the hyperthermophile *Pyrococcus furiosus*); and eukaryotes (such as *Saccharomyces cerevisiae*) [16].

Due to fundamental differences in cellular structure and genetic information, different electroactive microorganisms exhibit a high degree of diversity in terms of their electron transfer mechanisms and capacities. Thus, from the aspect of the direction of electron transfer, electroactive microorganisms can also be divided into electrogenic and electrophilic microorganisms. Electrogenic microorganisms, such as *Desulfovibrio* spp. [17] and *Geobacter* spp. [18], are capable of producing electrons during their metabolic processes and transferring these electrons to external electron receptors (e.g., oxygen, nitrate, sulfate, iron ions, and electrodes) for respiration [19]. Electrophilic microorganisms that have been applied in electrosynthesis, such as *Methanococcus* [20], *Acidithiobacillus ferrooxidans* [21], and *Clostridium* [22], can accept electrons from solid electrodes and then convert carbon dioxide into other valuable compounds in metabolism [23,24,25]. Although bacterial electrogenicity and electrophilicity are two opposing processes, certain electroactive microorganisms, such as *S. oneidensis* [26] and *G. sulfurreducens* [27], have the capacity to transfer electrons in both directions, thereby exhibiting the ability to act as both electrogenic and electrophilic bacteria.

These microorganisms possess a wide variety of metabolic pathways, allowing them to use both organic and inorganic molecules as sources of energy and carbon [28,29], and are capable of thriving under harsh environmental conditions, including hypoxic, hypersaline, and heavy metal-contaminated environments [30,31]. When these microorganisms come into contact with electrodes, their electron transfer efficiency as well as their growth and metabolism change in response to the electric field action [32]. These capabilities highlight the immense potential of electroactive microorganisms in applications ranging from energy production and environmental remediation to biosynthesis.

## 3. Electron Transfer Mechanism

Extracellular electron transfer (EET) describes the ability of microorganisms to associate their cellular metabolism with the electron flow in the surrounding environment [33]. The key to understanding the interaction mechanisms between electroactive microorganisms and electrodes is to clarify how this ability operates [34]. As illustrated in Figure 3A, two primary mechanisms of electron transfer can be distinguished at the microbe–electrode interface: direct electron transfer (DET) and indirect electron transfer (IET) [35].

Direct electron transfer (DET) is a prominent mode of electron transfer that requires direct physical contact between electrodes and microorganisms in the form of biofilms [32]. This contact demands that the microorganism possesses robust surface-adhesion capabilities and the ability to form stable biofilms [36]. DET can be further categorized into two distinct modalities based on the contact mode between electroactive microorganisms and electron acceptors (such as electrodes): short-range electron transfer, involving the cytochrome complexes located on the cell membrane [37]; long-range electron transfer, which, in contrast, involves cellular appendages such as conductive pili [38] or nanowires [39] that extend the range of the microorganism’s electron transfer capabilities. Taking the extensively studied model strains *G. sulfurreducens* and *S. oneidensis* as examples [40], their electron flows are schematically illustrated in Figure 3B. Some studies on *G. sulfurreducens* speculate that the electron flow begins with the inner membrane protein MacA, and then the electrons pass through the periplasmic protein PpcA to the outer membrane cytochromes (OMCs), such as OmcS, OmcB, and OmcZ [41]. Additionally, *G. sulfurreducens* produces conductive protein filaments, initially thought to be composed of PilA protein. However, later studies suggested that while PilA protein may play a bridging role in electron transfer, the primary conductive role is the cytochrome OmcS [42]. By culturing *G. sulfurreducens* in an electric field, Yalcin et al. proved that the electric field stimulated the production of OmcZ nanowires, with a conductivity 1000 times greater than that of OmcS nanowires [43]. It has been demonstrated that OmcZ is the principal type C cytochrome for facilitating DET between *G. sulfurreducens* and electrodes, with its eight heme groups providing a broad range of redox potentials (from −0.42 V to −0.06 V) [41]. Similarly, in *S. oneidensis*, the electron flow initiates at the inner membrane protein CymA, then transfers through the periplasmic protein FccA/STC to OMCs, and finally transfers electrons by direct contact with outer surface *c*-type cytochromes (such as OmcA and MtrC) to the extracellular electron acceptor [34]. In addition, *S. oneidensis* also produces nanowires, which are extensions of the bacterial outer membrane and periplasm containing cytochromes OmcA and MtrC, rather than pili [39]. It is clear that *c*-type cytochromes play an indelible role in the extracellular transfer of electrons.

**Figure 3 microorganisms-13-00631-f003:**
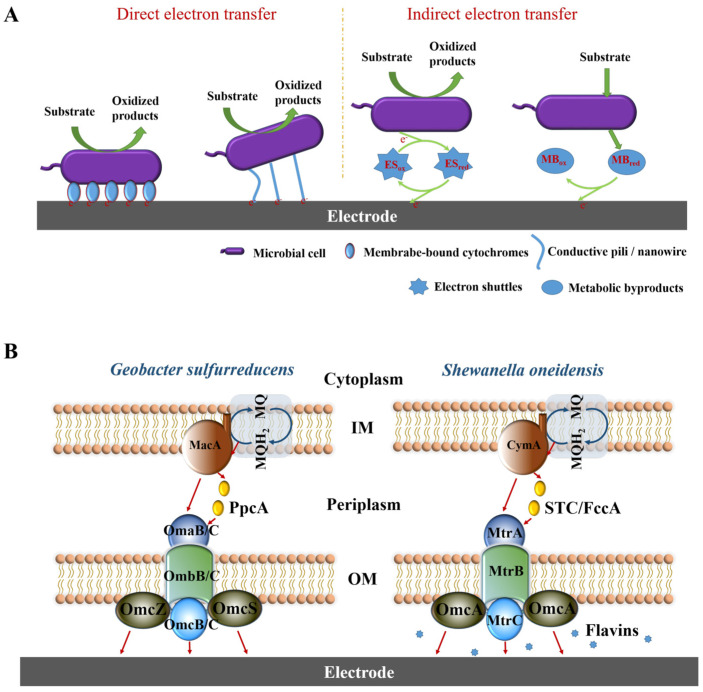
(**A**) Simplified representation of extracellular electron transfer mechanisms from microorganisms to the electrodes. (**B**) Electron transfer pathways in *G. sulfurreducens* and *S. oneidensis*, with the red arrow indicating the direction of electron transfer. Modified from [34,44].

Many electroactive microorganisms are capable of mediating electron transfer in a non-contact manner, known as indirect electron transfer (IET). This process primarily relies on two ways: one consists of mobile electron shuttles that can cycle through redox states, exemplified by flavins in *S. oneidensis* [45], phenazine derivatives in *Pseudomonas aeruginosa* [46], and quinones in *Lactococcus lactis* [47]. The other category involves metabolic byproducts secreted by bacteria themselves, such as hydrogen or formate, which constitute an irreversible redox reaction [34,48,49]. Many research efforts also aim to enhance the efficiency of extracellular electron transfer by genetically modifying electroactive microorganisms to produce more electron shuttles [50,51]. The logic behind these measures for IET efficiency improvement is to maximize the concentration as well as the diffusion capacity of electron shuttles in cellular secretions or the surrounding environment, thereby promoting effective collisions with electrodes [50].

Extracellular electron transfer has been the subject of extensive research [35,49,52]. This brief description is provided to facilitate a better understanding of the subsequent sections.

## 4. Effect of Anode Potential on Electricity Generation Behavior

The anode potential and the electrical current generation in MES are closely related to each other. The current is contingent upon the anode potential, representing the rate at which electrons are transferred to the electrode by electroactive microorganisms [53]. In these systems, electroactive microorganisms engage in oxidizing organic and inorganic substances through respiratory processes to generate electrons, reducing the terminal electron acceptors, and thereby harnessing energy. Generally, the redox potential of an electron acceptor determines the amount and ease of energy acquisition. Studies have shown that, compared to the redox potential of terminal proteins or mediators in the electron transfer chain, a greater value of positive anodic potential is thermodynamically more efficient for the process of transferring electrons from the cell to the receptor electrode through direct contact or chemical mediators [15]. This potential difference largely determines the driving force for electron transfer from microbial cells to the anode, as well as proton diffusion from a biofilm and the transport of hydroxide ions away from the cathode. The Nernst–Monod equation, a model that integrates the kinetics of microbial electrode reactions with electrode potential, can be used to characterize the relationship between the current density (*j*), which arises from the electron transfer reactions mediated by anode-respiring bacteria (ARB), and the anode potential (*E_anode_*) [54]. The Nernst–Monod equation takes the following form:(1)$$j=j_{max}\;[1+exp(-\eta F/RT)]$$
where *j* and *j_max_* denote the current density (A/m^2^) and the maximum current density, respectively; *F* and *R* are the Faraday constant (96,485 C/mol e^−^) and universal gas constant (8.3145 J/mol K), respectively; *T* is the absolute temperature (K); *η* represents the overpotential, determined by *η =* (*E_anode_* − *E_KA_*), where *E_KA_* is the anode potential at which the value of current density reaches *j*_max_/2. According to this equation, if the anode potential is greater than *E_KA_*, the accumulated ARB functions can achieve optimum effect in terms of generating high current density [54]. The Nernst–Monod equation also shows that within a certain potential range, the current density increases with anode potential until it reaches the *j*_max_ [54]. This finding highlights the critical role of anode potential in optimizing the performance of ARB within MESs to achieve enhanced bioelectricity generation.

The dependence of bioelectricity generation on anode potential certainly varies with species as well as anode materials. However, in bioelectrochemical systems, the ultimate goal of setting the anode potential is to develop the optimal microbial electrochemical capability for efficient electron transfer to the anode. As described in the following, this capability is typically reflected in a shorter start-up time and the ability to produce a high current density.

### 4.1. Effect of Anode Potential on Start-Up Time and Electric Current/Power Density

In MES, the start-up time and the maximum current/power density are two critical performance indicators. The start-up time usually refers to the duration required for MFCs to transition from initiation to a stable operational state. During this phase, electroactive microorganisms undergo a series of processes, including environmental acclimatization, biofilm formation, microbial community development, electron transfer pathways establishment, and stabilization of cellular performance. Consequently, this period is influenced by many factors, such as system configuration, inoculum composition, substrate type, and external resistance [55]. A stable working state does not emerge instantaneously and may range from several hours to several months [56]. The maximum current/power density denotes the highest current flow/electrical energy output per unit area of the anode achievable during MFCs operation, reflecting the electron transfer capacity of the biofilm and the power generation efficiency of the cell. For practical applications, reducing the start-up time and enhancing the maximum current/power density are key objectives in improving the efficiency and practicality of MFCs.

Studies have shown the significant impact of anode potential on the performance of MESs, particularly in terms of the start-up time and the maximum current density [57,58,59]. For example, Wang et al. applied a balanced anode potential of 0.2 V in a two-chamber microbial fuel cell (MFC) and observed a reduction in start-up time from 59 days to 35 days [57]. During this period, the current output increased from 0.42 mA to 3 mA due to the increased driving force of substrate oxidation. So, they achieved a 41% reduction in start-up time and a 614.3% increase in current output at +0.2 V in a two-chamber MFC. These findings indicate that optimizing anode potential can significantly enhance the electrochemical activity of the anodic biofilm during the start-up phase, thereby shortening the start-up time and increasing current density [57]. By exploring a broader spectrum of anode potentials, Torres et al. discovered that microbial electrolytic cells exhibited a more rapid increase in current density at anode potentials of −0.15 V and −0.09 V compared to that at +0.37 V. This is because ARBs using solid conductive substrates (such as *G. sulfurreducens*) are more likely to form continuous connections on the anode surface at low anode potentials, reducing energy loss through DET, thus promoting rapid start-up [58]. Buitrón et al. further pointed out that optimizing the external resistance can also enhance the growth of electroactive bacteria, thereby reducing the start-up time from 8 days to 4 days (a decrease of 50%) while producing higher current and power densities [59].

Taking the start-up time and the maximum current/power density as performance indicators, we summarize recent studies on the influence of anode potential on the electricity production capacity of electroactive microorganisms—categorized by the model strains *Geobacter* species, *Shewanella* species, and ARB communities in Table 1. We can see that there is no singular set potential that always universally guarantees superior outcomes, which depends on the culture conditions, electrode material, inoculum, the structure of the electrochemical system, and other factors. Of the 22 studies listed, 82% indicated that a more positive anodic potential for a given experimental anodic potential can increase the protein expression level and metabolic activity related to electron transfer and electrochemical reaction kinetics. This, in turn, leads to increased energy yield. Meanwhile, 23% considered 0 V as the optimal potential while just 5% suggested that a more negative potential could be advantageous, particularly in electroactive microbial communities. More positive potentials favor the growth of electroactive microorganisms capable of directly transferring electrons via solid conductive matrices. These microorganisms use minimal energy to grow and contribute to high current density, which is crucial for the desired output of bioelectrochemical systems. Overall, these studies show that higher potentials generally enhance performance. However, these high potentials were only studied up to about 0.8 V, which may be considered low in the context of different experimental designs. Based on these findings, we hypothesize that in most cases, a high anodic potential is likely to provide higher performance for microbial electrochemical systems, provided there is no cell damage. Identifying the critical threshold for cell damage is imperative for the establishment of the optimal experimental condition. Additionally, these studies also suggest that start-up time and maximum current/power density are often simultaneously affected by anode potential. This is because the anode potential not only fosters the metabolic activity of electroactive microorganisms but also modulates their electron transfer pathways and the kinetics of electrochemical reactions, thereby optimizing the entire process [57].

### 4.2. Effect on the Electron Transfer Pathways of Microorganisms

The energy yield from bacterial respiration is determined by the thermodynamic energy between the electron donors and the electron acceptors, as well as the efficiency of their respiratory process [58]. The applied anode potential can alter the electron affinity of the electron acceptors, modify the transmembrane electrochemical gradient of the proton transfer processes, and change the kinetics of electron transfer between electron carriers, which leads to the increase in EET rate, the NAD+/NADH ratio, and the utilization of the thermodynamic frame (i.e., maximum available energy from substrate oxidation) [64]. In this process, electroactive microorganisms modulate the composition and activity of the electron transfer components and switch their respiratory mechanism to adapt to varying anode potentials, thereby maximizing energy gain.

Taking *Geobacter* species as an example, they encode a large number of *c*-type cytochromes. However, only a few cytochromes are characterized and participate in energy metabolism [79]. These cytochromes are distributed in subcellular locations, including the inner membrane (Cbc complex, ImcH, MacA), periplasm (PpcA homologs, PccH), outer membrane (porin–cytochrome complex, OmcS, OmcZ, and other membrane *c*-type cytochromes) [80], and enable *Geobacter* to employ multiple electron transfer paths [81,82,83]. The electrode potential may simulate the existence of different electron receptors, thus inducing specific electron transfer pathways. For instance, Zhu et al. revealed that *G. sulfurreducens* expressed different EET components under five different anode potentials (−0.46 V, −0.3 V, 0 V, 0.3 V, and 0.6 V). Using cyclic voltammetry (CV) and first derivative cyclic voltammetry (DCV), they discerned that biofilms acclimated under −0.30 V, 0 V, and 0.30 V shared analogous electric spectra, yet distinct from those developed under −0.46 V and 0.60 V. Intriguingly, the biofilms acclimated to an anode potential of −0.46 V displayed oxidation peaks at −0.48 V and −0.42 V, while those acclimated to 0.6 V exhibited peaks at −0.16 V and 0.58 V. These findings underscore the profound influence of anode potential on the microbial electron transfer mechanisms [65]. Given the midpoint potentials of common cytochromes in *Geobacter*, which are −0.42 V for OmcZ [79], −0.39 V for OmcB [84], and −0.37 V for the periplasmic cytochrome PpcA [85], the emergence of novel oxidation peaks at anode potentials of −0.46 V and 0.6 V significantly broadens the spectrum of responses exhibited by *Geobacter* biofilms to anodic potential variations. Furthermore, Busalmen et al. investigated the polarization effects on *G. sulfurreducens* at 0.6 V and 0.1 V, revealing that biofilms subjected to different anode potentials exhibited entirely distinct redox couples [60]. The manifestation of these novel redox couples is hypothesized to arise from structural transformations of the cytochrome proteins in response to changes in electrode potential. These transformations are thought to facilitate efficient electron exchange with the electrode surface. This adaptive mechanism likely represents a bacterial strategy to optimize electron transfer efficiency in accordance with varying electrode potentials.

Compared with *Geobacter* species that predominantly conduct DET, *Shewanella* species utilize both direct and IET mechanisms and thus can be affected by anode potential in a richer way. Peng et al. investigated the impact of electrode potential on the bioelectricity generation of *S. oneidensis* and found that the anode potential could regulate the accumulation of cell surface cytochromes OmcA/MtrC [86]. They observed that *c*-type cytochromes were not expressed at a surface potential of −0.24 V. However, when the potential was switched to 0 V, *c*-type cytochromes enriched at the bacteria–electrode interface, leading to a sudden increase in current followed by a gradual decrease over time. This phenomenon may be attributed to the accumulation of cytochromes hindering the more efficient electron transfer of flavins involved in IET [86]. Moreover, Grobbler et al. observed that as the voltage increased, there was a significant upregulation in the expression of key EET proteins (including MtrABC, OmcA, and riboflavin synthase) [66]. Additionally, elongation factor FusB and ribosome maturation factor RimM, which are involved in protein synthesis, were also upregulated. These changes contributed to an enhanced abundance of EET mediator synthesis proteins, which led to a significant increase in maximum current density [66]. Additionally, TerAvest et al. reported that *S. oneidensis* achieved the peak current density at a moderate anode potential of 0.397 V, with a subsequent decline in current and coulomb efficiency at an even larger positive potential. This reduction was not attributed to a general cellular stress response, but rather to the direct damaged electron transfer proteins, such as *c*-type cytochromes, on the electrode surface under the more positive potential [68].

Together, these studies illustrate that anode potential can regulate the selection, expression, or structural reconstruction of electron transfer-related genes and proteins in microorganisms, thereby influencing the electron transfer process (Figure 4). However, while a larger value of positive anode potential is generally advantageous for electron transfer from bacteria to electrodes, it is also crucial to select an optimal anode potential that is below the damage threshold of the electron transfer proteins to prevent their degradation.

### 4.3. Effect on Microbial Community

Natural environments such as wastewater often harbor a rich diversity of electroactive microorganisms, each capable of utilizing distinct substrates and metabolic pathways for electron transfer. This metabolic diversity fosters the formation of synergistic or competitive interactions among different species, which in turn influences the collective function and electron transfer efficiency of the microbial community. Meanwhile, a diverse microbial community can enhance the system’s stability and adaptability to fluctuating environmental conditions, ensuring consistent performance. Consequently, a higher relative abundance of these electroactive microorganisms is advantageous for achieving higher maximum power density and current density [87].

The magnitude of the applied anode potential can selectively enrich electroactive microorganisms with largely different energy harvesting abilities, thereby impacting the diversity index of the microbial community [64,88,89,90,91]. For example, Zhao et al. utilized glucose as a model substrate to investigate the impact of different anode potentials (−0.15 V, 0 V, and 0.2 V) on current generation, electron transfer pathways, and the abundance of electricity-producing microorganisms within a two-chamber H-type MES reactor with a working volume of 250 mL [87]. Their findings revealed that elevated anode potentials (0 V and 0.2 V) can modulate the degradation pathways of complex substrates within MESs, thereby enhancing current generation from 0.07 mA/cm^2^ to 0.12 mA/cm^2^ and coulombic efficiency from 11% to 22% during microbial domestication. Moreover, they observed a direct correlation between the magnitude of the current generation and the abundance of microbial communities in the anode biofilm. The abundance of the microbial community in the anode biofilm increased by 29% when the anode potential was raised from −0.15 V to 0 V, and by 39% when increased to 0.2 V [87]. In addition, Ishii et al. employed a dynamic metatranscriptomic approach to identify the complex microbial communities’ responses to distinct electron transfer stimuli within MFCs, specifically under basic MFC conditions, 0.1 V constant potential (which enhances the electron transfer rate), and open circuit conditions (which decreases the electron transfer rate). Their findings revealed that the current density, measured at an anode potential of 0.1 V, reached 550 mA/m^2^, which is eightfold greater than that of the baseline MFC [92]. Furthermore, the study identified two microbial groups, *Desulfobulbaceae* and *Desulfuromonadales*, that exhibited significant gene expression responses under the influence of EET stimuli [92].

In MESs, high diversity of microbial communities typically signifies the presence of multiple electron transfer mechanisms and a broad range of substrates that can be utilized. This diversity allows MESs to be more resistant to external interference (such as changes in toxins and nutrients) under the complementary action of different microorganisms during long-term operation [93]. This resilience is instrumental in maintaining the stability of the energy conversion process. From the aforementioned context, it can be inferred that an appropriate anode potential can promote the diversity of the microbial community and enhance the in situ activity of electricity-producing bacteria, thus better realizing the functionality and stability of the MESs.

### 4.4. Summary of Anode Potential Effects on Electricity Generation Behavior

The anode potential serves as a master variable in microbial electrochemical systems, intricately shaping electron transfer efficiency and community dynamics. A summary of key findings is presented in Table 2. The majority of research indicated that a more positive anodic potential can increase current density and reduce start-up time. However, the optimal anode potential varies depending on the specific microbial strains, electrode materials, and system configurations. For instance, *Geobacter* species tend to perform better at lower potentials, while *Shewanella* species show higher activity at moderate potentials. But mechanistically, the influence of anode potential on the electron transfer paths and the dynamic change in microcolonies is similar in these two cases. From the perspective of the electron transfer path, as the anode potential rises, electroactive microorganisms undergo several adaptive changes to optimize energy conversion and metabolic processes, thus boosting extracellular electron transfer (EET) efficiency. These changes entail an increase in the abundance of EET proteins and factors involved in protein synthesis, the enrichment of *c*-type cytochromes on the electrode surface, and the generation of new configurations of *c*-type cytochromes or the selection of novel EET paths. However, when the anode potential is too high, the structural integrity of *c*-type cytochromes may be impaired, and the expression of genes related to EET may be further modified, decreasing their EET capacity. Another aspect from which the effects of anode potential on electricity generation behavior can be reflected is the microbial community. Elevated anode potentials could select microorganisms with electrochemical activity and increase the abundance of microbial communities in the anode biofilm.

## 5. Effect of Anode Potential on Physiology

The physiological behaviors of microorganisms involve a complex set of biochemical processes and functions that are pivotal for microorganisms to survive and multiply in nature. The application of anodic potential has a significant impact on electroactive microorganisms, influencing not only the aforementioned electricity-generating capabilities but also a range of physiological behaviors, including mobility, adhesion, metabolism, and biofilm formation (Figure 5).

### 5.1. Effect on the Mobility of Microorganisms

Microbial movement behavior, a key physiological attribute, is facilitated by specific appendages such as flagella or pili, enabling microorganisms to navigate their environment in search of nutrients or to evade adverse conditions. While anode potential serves as a critical regulatory factor, modulating the motility and chemotaxis of electroactive microorganisms, its effects are context-dependent and often contradictory across studies, raising questions about underlying mechanisms and ecological implications. For example, Grobbler et al. discovered that *S. oneidensis* cultured at −0.19 V exhibited an increased abundance of proteins associated with chemotaxis and motility compared to cultures at higher anode potentials of +0.71 V and +0.21 V, suggesting energy-limited conditions drive exploratory behavior [66]. Conversely, Harris et al. reported “electrokinesis”—a transient surge in *S. oneidensis* swimming speed near electrodes at potentials from −0.6 V to +0.6 V—which paradoxically coexists with reduced chemotaxis protein abundance at higher potentials [67]. This discrepancy implies that motility regulation under anode potential may involve redox-sensitive signaling pathways independent of protein synthesis, yet such mechanisms remain unverified. In addition, Berthelot et al. investigated the effect of an applied electrical current (0, 0.07 mA, 0.125 mA) on bacterial flagella-mediated motility [94]. They found that applied electric currents decreased the motility of *Escherichia coli* and *Pseudomonas aeruginosa* while enhancing their directional movement. However, the impact of anodic current varied among different bacterial species. For instance, *Escherichia coli* showed a decreased average speed under all conditions, while *P. aeruginosa* exhibited a significant decrease in cell speed only at 0.07 mA [94]. These differences likely stem from variations in surface charge, membrane composition, or flagellar architecture, but no study has systematically correlated bacterial surface properties with motility responses to anode potential. The aforementioned studies have delved into the influence of anodic potential on flagella-mediated swimming motility, which is of great importance for understanding how electroactive microorganisms interact with electrodes in MESs and how electron transfer affects bacterial movement. However, most studies rely on endpoint protein assays or bulk motility measurements, failing to resolve real-time behavioral dynamics at the electrode–biofilm interface. For example, electrokinesis was only observed in cells closest to electrodes [67], suggesting spatial heterogeneity in motility regulation that conventional methods overlook. Advanced tools like in situ electrochemical microscopy or single-cell tracking could bridge this gap but are rarely applied. Additionally, pilus-mediated twitching motility and secretion-directed chemotaxis are theoretically also influenced by the electrical or chemical gradients generated by electrodes [95,96]. Conductive pili in *Geobacter* facilitate both electron transfer and surface attachment, suggesting a dual role that may create trade-offs between motility and electroactivity under varying potentials. Those remain largely unexplored but significantly affect the bacterial ability to adapt to their surroundings and form biofilms.

### 5.2. Effect on the Adhesion and Growth of Microorganisms

Surface adhesion is another critical behavior for microbial survival and biofilm functionality. The electrochemical activity of biofilms is intricately linked to the layer of cells that are physically directly attached to the electrode surface, which acts as a form of “electrochemical gate” modulating the electron transfer between the biofilm and the electrode surface [62]. Optimal electrode potentials can facilitate the adhesion of microbial cells on the electrode surface, providing more energy for microorganisms and influencing their growth rate and reproductive ability, which are key factors in electrochemical systems. For example, Busalmen et al. observed that increasing polarization potential from 0.1 V to 0.6 V promoted the adhesion of negatively charged *G. sulfurreducens* on the electrode surface due to electrostatic interaction [60]. However, it is important to consider that the relationship between electrode potential and microbial adhesion is not universally applicable and may vary depending on the specific microorganisms and environmental conditions. For the mixed microbial communities, lower anode potentials (−0.15 V and −0.09 V) are more favorable for enriching electroactive microorganisms that can utilize solid conductive matrices for DET. These microorganisms achieve optimal growth with minimal energy expenditure. In contrast, a high anode potential (0.37 V) favors the growth of non-electroactive microorganisms or electroactive microorganisms using electron shuttle molecules [58]. This dichotomy raises concerns about the universality of “optimal” potentials, as real-world systems often require balancing electroactivity with community diversity for long-term stability. Additionally, externally applied electric fields can modulate cellular developmental processes. For instance, research on *Pseudomonas fluorescens* demonstrated that an anodic potential of −0.2 V enhanced the bacterial growth rate, which is evidenced by the increased cell length and reduced doubling times compared to those observed under an anodic potential of +0.5 V. These effects are often linked to localized disruptions in the cell membrane’s electrochemical potential [97]. Yet, such studies rarely address whether these effects stem from specific ion flux alterations or broader metabolic reprogramming. Furthermore, the external electric field can impact cellular conductive appendages. For example, Yalcin et al. found that applying an electric field to *G. sulfurreducens* biofilms induced overexpression of OmcZ, resulting in the production of OmcZ nanowires (~2.5 nm diameter filaments) with 1000-fold higher conductivity than OmcS nanowires (~3.5 nm diameter filaments) [43]. While this highlights adaptive conductive appendage remodeling, the energy cost of nanowire production and its trade-offs with other cellular functions (e.g., motility, substrate uptake) remain unquantified. Current investigations into the effects of anodic potential on the adhesion and growth of electroactive microorganisms remain inadequately systematic, with many studies relying on extrapolated conclusions derived from assumptions. To address these limitations, the integration of multivariate optimization strategies and advanced in situ metabolomics profiling emerges as a promising approach to elucidate the intricate relationship between defined potential gradients and adhesion-growth dynamics. Such methodologies could provide mechanistic insights into energy reallocation patterns and biofilm–electrode interfacial adaptations under electrochemically modulated conditions, thereby bridging critical knowledge gaps in the field of electrochemical–microbial interactions.

### 5.3. Effect on the Metabolism of Microorganisms

Studies have shown that the anode potential can change the metabolic path, thus affecting the electron transfer efficiency. For example, Atsumi Hirose et al. [72] demonstrated that for *S. oneidensis* MR-1, higher electrode potentials (+0.5 V, +0.2 V, and 0 V) facilitated current generation by selecting metabolic pathways under certain anode potentials. Under medium and low electrode potentials, *S. oneidensis* MR-1 primarily relies on the pyruvate formate-lyase (PFL)-dependent formate pathway for energy production. In contrast, at high electrode potentials, the bacteria utilize an NADH-dependent metabolic pathway involving pyruvate dehydrogenase (PDH) and NADH dehydrogenase (NDH) to oxidize pyruvate [72]. The Arc regulatory system in MR-1 is implicated in sensing electrode potentials and modulating the expression of catabolic genes. Activation of the NADH-dependent pathway at high anode potentials increases the proton motive force (PMF) generated per electron, thereby enhancing ATP production [72]. This metabolic adjustment may also serve as a protective mechanism, allowing the bacteria to adapt to varying environmental pressures by modulating their metabolic pathways. This adaptive flexibility is crucial for their survival in diverse electrochemical environments and holds significant implications for the optimization of microbial electrochemical systems. In addition, as a fundamental metabolic pathway for energy production and biosynthesis in electroactive microorganisms, the tricarboxylic acid cycle (TCA) is also significantly influenced by the anodic potential. For instance, Grobbler et al. observed the TCA cycle of *S. oneidensis* MR-1 at three distinct electrode potentials (+0.71 V, +0.21 V, −0.19 V) and discovered that there were branches at varying anode potentials [70]. At higher anode potential (+0.71 V), enzymes related to the TCA cycle were more abundant, indicating increased metabolic activity. Conversely, Matsuda et al. found that the activity of TCA cycle activity in *S. Loisica* PV-4 cells could be reversibly regulated by changing the anode potential (0 V, 0.4 V) [71]. However, this activity showed an inverse relationship with current and electron acceptor energy levels. Specifically, the TCA cycle was activated at low potential (0 V) and deactivated at high potential (0.4 V). This unique behavior is hypothesized to stem from the internal redox state inside the cells becoming oxidative due to excessively efficient EET, triggering an antioxidant stress response that deactivates certain components of the TCA cycle, such as Complex II [71]. It is important to note that these studies employed distinct experimental setups, including different bacterial strains, electrode materials, and electrolytes. Consequently, the 0.4 V anode potential referenced in the latter study may impose higher stress via excessive EET efficiency on *Shewanella* species compared to the 0.71 V anode potential in the former study. Furthermore, Nakagawa et al. investigated the metabolic traits of an engineered *S. oneidensis* strain, introducing glycolytic genes under electrode-respiring conditions [73]. They discovered that the anode potential could alter the carbon metabolic pathways of *S. oneidensis*. Specifically, when the anode potential was set at +0.4 V, the engineered strain oxidized glucose to acetate, producing D-lactate as an intermediate metabolite. In contrast, no accumulation of D-lactate was observed when the strain was grown at an anode potential of 0 V. Additionally, the transcription of genes related to pyruvate and D-lactate metabolism was upregulated at higher anode potentials compared to lower ones, indicating a regulatory role of anode potential in the metabolic activities of *S. oneidensis* [73]. In addition to *Shewanella*, comprehensive metabolomic analyses of *Geobacter* have similarly demonstrated that the anode potential exerts a significant influence on cellular metabolism [98]. Specifically, cells cultivated on glassy carbon anode poised at +0.2 V exhibited elevated concentrations of TCA cycle metabolites (such as adenosine triphosphate/adenosine diphosphate ratio) compared to those cultured at −0.2 V. Collectively, these studies underscore the pivotal role of anode potential in modulating the metabolic pathways and electron transfer efficiency of electroactive microorganisms. The ability to sense and respond to anode potentials allows these bacteria to optimize their metabolic strategies for energy production and electron transfer. This adaptive flexibility is crucial for their survival in diverse electrochemical environments and holds significant implications for the optimization of microbial electrochemical systems.

### 5.4. Effect on the Formation of Electroactive Biofilms

Microbial biofilms are structurally defined as microbial communities embedded within a matrix of extracellular polymeric substances (EPS), which include polysaccharides, proteins, extracellular DNA, and biosurfactants [99]. Electroactive biofilm is a key component of the MESs that realizes electron transfer through interaction with electrodes. For some bacteria, electrical activity depends not on cell density, but on biofilm formation [100,101]. In conductive biofilms, where every cell contributes to electron transfer, the thickness of the biofilm directly correlates with current or product generation, with multilayer biofilms generally outperforming single-layer biofilms [102,103]. Thus, biofilms with strong surface adhesion and robust biofilm-forming properties are crucial for the application of bioelectrochemical systems [34]. As a controllable parameter, the anode potential plays a pivotal role in modulating the electroactive biofilms. Studies have shown that both current density and biofilm biomass are positively correlated and increase concurrently as the electrode potential is within a certain range [32]. When the anode potential exceeds a certain threshold—such as 0.397 V in *S. oneidensis* [68] and 0.21 V in wastewater inoculum [76]—both biomass and current density gradually decline with further increases in anode potential. Furthermore, the applied mode of the anode potential and the magnitude of the external resistance also influence the flatness and compactness of biofilms. Specifically, continuous polarization tends to produce relatively flat biofilms, whereas periodic polarization leads to the formation of mushroom-like structures on the upper layer of biofilms [104]. Moreover, higher external resistance (>50 Ω) resulted in compact biofilms, while lower resistance (<50 Ω) produced looser biofilms [105]. These findings suggest that the physical characteristics of biofilms can be manipulated through the control of anode potential and external resistance, but the long-term stability and functionality of these biofilms under varying conditions need further investigation. In addition to structure and biomass, the EPS components of electroactive biofilms play complex roles in electron transfer [106,107,108,109]. A study on *Shewanella* species exhibited that the non-conductive extracellular polysaccharides in biofilms formed at 0.4 V were significantly lower than those at 0 V, which contributed to the electron transfer efficiency [69]. Conversely, in *Geobacter* species, biofilms cultivated at −0.2 V and 0 V produce more extracellular redox-active proteins but less extracellular polysaccharide, while, those at 0.4 V and 0.6 V contain a non-conductive inner layer of extracellular polysaccharide that may impede DET [61]. The observed differences may be attributed to the distinct metabolic pathways and electron transfer mechanisms that characterize *Geobacter* species in contrast to those of *Shewanella* species, resulting in *Geobacter* species activating the protection mechanism to secrete more polysaccharide under identical voltage (e.g., 0.4 V). These contrasting findings highlight the species-specific responses to anode potential, emphasizing the need for a tailored approach to biofilm optimization based on the microbial species involved. Furthermore, the influence of anode potential on EPS of mixed community biofilms is more complicated. For instance, in artificial brewery wastewater cultivated at different anode potentials (−0.3 V, 0 V, +0.3 V, +0.6 V), the current magnitude and the redox-active humic substances in EPS fractions were highest at 0 V and lowest at +0.6 V, while non-conductive polysaccharide was highest at +0.6 V and lowest at 0 V [77]. This non-linear relationship between EPS content and anode potential is attributed to the mixed microbial community, where different species may respond differently to the applied potential, leading to complex interactions and outcomes. Overall, the biomass, structure, and EPS composition of electroactive biofilms under different anode potentials are regulated by cells to maintain a balance between EET functionality and cellular protection [61,77,106]. However, the mechanisms underlying these regulatory processes are not fully understood and require further research to develop a comprehensive understanding of biofilms in MESs.

### 5.5. Summary of Anode Potential Effects on Microbial Physiology

Electroactive microorganisms exhibit dynamic physiological adaptations under varying anode potentials, influencing their motility, adhesion, metabolism, and biofilm formation. Reported studies showed that applied anode potentials can modulate motility (promoting or impeding swimming on specific conditions), adhesion (selecting species, reducing cell size/growth rate, and inducing conductive appendages under high potentials), and metabolism (enhancing activity and pathway selection, though inhibited at extremes). Biofilm biomass and structure align with potential levels, balancing conductive and non-conductive extracellular polymers. These responses are critical for optimizing microbial electrochemical systems (MESs). A summary of key findings is presented in Table 3.

## 6. Conclusions and Future Outlook

This review provides a summary of the progress in understanding the role of anode potential in electromicrobiology, with a focus on how anode potential impacts electricity generation and the physiological behaviors of electroactive microorganisms. The diverse species of electroactive microorganisms and their varied metabolic and EET pathways determine the complexity of the anode potential’s influence. In the context of the entire MES, the anode potential serves as the driving force for electrons and protons, directly affecting the rate of EET and, consequently, the magnitude of the current. However, for the electroactive microorganisms themselves, factors such as the redox activity of proteins in their metabolic pathways and the EET pathways, and the development of biofilms determine that the effect of anode potential is non-monotonic, i.e., there is an optimal range of potential values for MES performance. By optimizing anode potential, the performance of MESs can be significantly enhanced, reducing start-up time and increasing maximum current/power density. The comprehensive understanding of the impact of anode potential on electricity generation and the physiological behaviors of electroactive microorganisms provides a scientific basis for the optimization of microbial electrochemical systems and their applications in various fields.

The development of efficient and stable MESs is currently an important research direction in the fields of environmental science and energy engineering. Through these studies, we have gained a better understanding of the interaction mechanisms between electroactive microorganisms and anodic potential, providing theoretical and technical support for the advancement of microbial electrochemical technology. These insights not only help improve the performance of MESs but also hold potential for breakthroughs in bioenergy production, environmental remediation, and biosensor development. However, this field also faces challenges, including achieving precise control of the potential to adapt to the dynamic changes in microbial communities and enhancing the stability and reproducibility of the system. In addition, factors such as the selection of electrode materials, the optimization of microbial strains, and the regulation of operating conditions also impact the effect of potential regulation. Future research will continue to explore the mechanism of potential modulation in MESs and to optimize microbial physiological activities and electro-production behaviors through potential modulation. The development of novel electrode materials and modification techniques will improve the precision and response rate of potential modulation. Meanwhile, synthetic biology and systems biology approaches will be used to modify and optimize microbial strains to enhance their adaptability to potential changes and efficiency of electricity production. Here, we propose a “Smart Voltage-Responsive Communities” framework (Figure 6), where electrode materials are innovated to achieve corrosion-resistant and highly conductive activity, synthetic biology tools enable the design of voltage-responsive microbial strains, while systems biology deciphers metabolic–electrochemical coupling under varying potentials. For instance, electrode materials integrate bioresponsive interfaces (e.g., graphene–cytochrome hybrids) to synchronize potential modulation with microbial activity. Embedding voltage-sensitive promoters into electroactive microorganisms could dynamically regulate cytochrome expression, achieving adaptive electron transfer. Additionally, establishing a voltage gradient forms spatially stratified communities (such as high-potential zones and low-potential zones). Concurrently, machine learning models trained on multi-omics datasets may predict optimal potential windows for specific microbial communities. Ultimately, this framework aims to bridge lab-scale discoveries and industrial applications, driving the development of next-generation MESs that are both efficient and scalable.

## Figures and Tables

**Figure 1 microorganisms-13-00631-f001:**
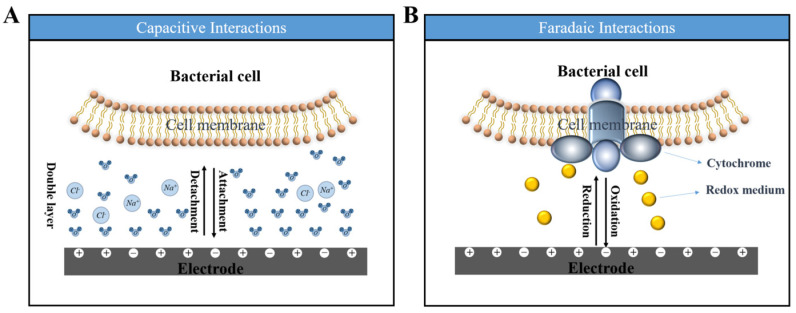
(**A**) Capacitive interactions and (**B**) Faradaic interactions between microorganisms and electrodes. Modified from [1].

**Figure 2 microorganisms-13-00631-f002:**
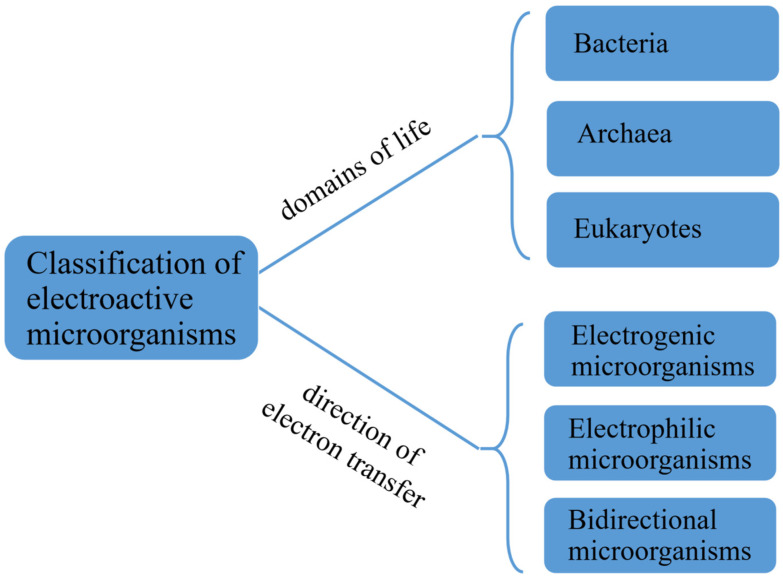
Classification of electroactive microorganisms.

**Figure 4 microorganisms-13-00631-f004:**
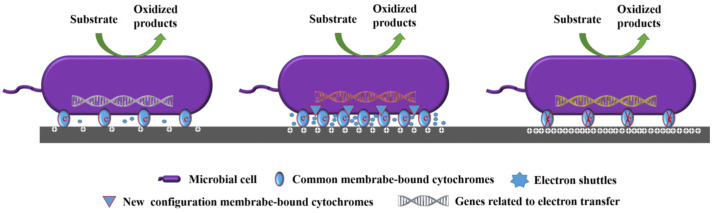
Influence of anode potential on EET pathways in electroactive microorganisms. i. Under low or zero applied potential conditions, electroactive microorganisms perform normal EET, transferring electrons from the intracellular environment to electrodes through their inherent mechanisms. ii. As the anode potential increases, several adaptive changes occur in electroactive microorganisms to optimize energy conversion and metabolic processes, thereby enhancing EET efficiency. These changes include the increase in the abundance of EET proteins and factors involved in protein synthesis; the enrichment of *c*-type cytochromes on the electrode surface; the increase in the secretion of soluble electron shuttles for IET; the generation of new configurations of *c*-type cytochromes or the selection of new EET paths. iii. When the anode potential is excessively high, the structural integrity of *c*-type cytochromes may be compromised, and the expression of genes related to EET may be further altered, thereby reducing their EET capacity.

**Figure 5 microorganisms-13-00631-f005:**
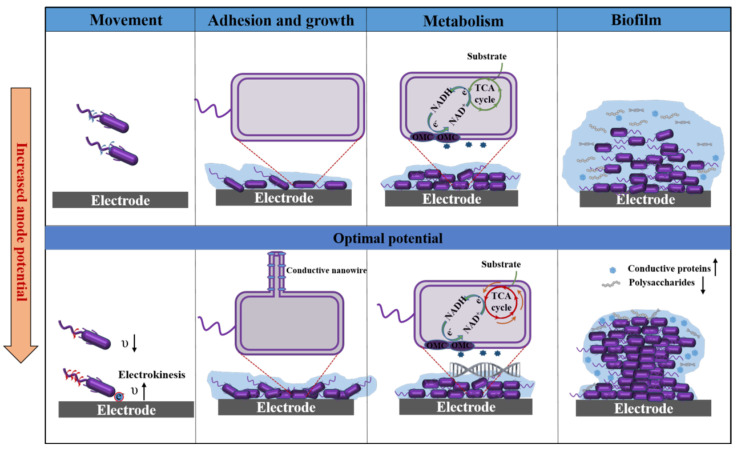
The influence of anode potential on physiology of electroactive microorganisms: i. Elevating the anode potential significantly reduces the abundance of proteins related to bacterial chemotaxis and mobility but enhances the swimming speed during transient electrokinesis. ii. Elevating the anode potential significantly shortens cell length, extends the doubling time, and induces the formation of conductive nanowires. iii. A moderate increase in anode potential can enhance metabolic activity and choose a more suitable metabolic pathway. iv. A moderate increase can promote biomass accumulation, decrease the total amount of extracellular polymeric substances (EPS), and increase the conductive components within EPS. Arrows pointing upward or downward indicate an increase or decrease in speed or EPS.

**Figure 6 microorganisms-13-00631-f006:**
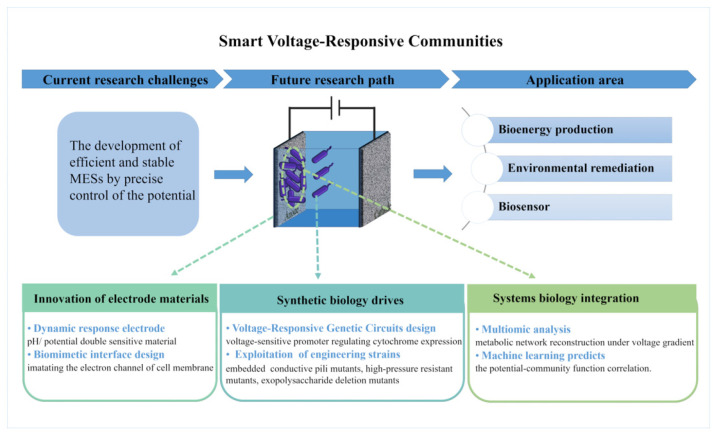
Proposed framework for advancing microbial electrochemical systems (MESs) through synthetic biology, systems biology, and potential modulation.

**Table 1 microorganisms-13-00631-t001:** Applied potential studies comparing set anode potential.

Inoculum	Applied Potential (V)	Working Electrode	Reference Electrode	Optimal Potential (V)	Ref.
*G. sulfurreducens*	0.1, 0.4, 0.6 vs. Ag/AgCl	Graphite felt	Ag/AgCl	0.6	[60]
*G. soli*	−0.4, −0.2, 0, 0.2, 0.4, 0.6 vs. SCE	Graphite plate	SCE	0, 0.2	[61]
*G. sulfurreducens*	−0.2, −0.1, 0, 0.2 vs. Ag/AgCl	Stainless steel	Ag/AgCl	0.2	[62]
*G. sulfurreducens*	−0.16, 0, 0.4 vs. SHE	Carbon paper	SCE	0, 0.4	[63]
*G. sulfurreducens*	−0.2, −0.1, 0, 0.2, 0.4, 0.6 vs. SHE	——	——	≥0.2	[64]
*G. sulfurreducens*	−0.46, −0.3, 0, 0.3, 0.6 vs. Ag/AgCl	Graphite plate	Ag/AgCl	0, 0.3, 0.6	[65]
*S. oneidensis*	−0.19, 0.21, 0.71 vs. SHE	Carbon cloth	Ag/AgCl	0.71	[66]
*S. oneidensis*	−0.6, 0, 0.3, 0.6 vs. graphite	Graphite fiber	graphite	0.6	[67]
*S. oneidensis*	−0.003, 0.197, 0.397, 0.597, 0.797 vs. SHE	Graphite paper	Ag/AgCl	0.397	[68]
*S. putrefaciens*	−0.1, 0, 0.1, 0.2, 0.3, 0.4 vs. Ag/AgCl	Polycrystalline carbon rod	Ag/AgCl	0.4	[32]
*S. oneidensis*	0, 0.4 vs. SHE	Graphite plate	Ag/AgCl	0.4	[69]
*S. oneidensis*	−0.19, 0.21, 0.71 vs. SHE	Carbon cloth	Ag/AgCl	0.71	[70]
*S. loihica*	0, 0.4 vs. SHE	ITO-coated glass	Ag/AgCl	0	[71]
*S. oneidensis*	0, 0.2, 0.5 vs. SHE	Graphite felt	Ag/AgCl	0.5	[72]
Engineered *S.* *oneidensis*	0, 0.4 vs. Ag/AgCl	Graphite felt	Ag/AgCl	0.4	[73]
*S. oneidensis*	0, 0.2, 0.35, 0.5 vs. Ag/AgCl	Graphite plate	Ag/AgCl	0.5	[74]
Primary clarifier effluent	−0.46, −0.24, 0, 0.5 vs. Ag/AgCl	Carbon fiber brush	Ag/AgCl	0.5	[75]
Primary clarifier effluent	−0.25, −0.09, 0.21, 0.51, 0.81 vs. SHE	Graphite plate	Ag/AgCl	0.21	[76]
artificial brewery wastewater	−0.3, 0, 0.3, 0.6 vs. SCE	Graphite plates	SCE	0	[77]
ARB communities	−0.15, −0.09, 0.02, 0.37 vs. SHE	Graphite rod	Ag/AgCl	−0.15	[58]
Domestic wastewater	0, 0.2 vs. Ag/AgCl	Graphite	Ag/AgCl	0.2	[57]
Marine sediment	−0.058, 0.103, 0.618 vs. Ag/AgCl	Graphite	Ag/AgCl	0.618	[78]

**Table 2 microorganisms-13-00631-t002:** Summary of anode potential effects on electricity generation behavior.

Electricity Generation Aspect	Key Findings	Ref.
Start-up time and electric current/power density	A more positive anodic potential can increase current density and reduce start-up time.	[60,66]
Electron transfer pathways	Electrode potential induces structural transformations of cytochrome to form specific electron transfer pathways, thereby enhancing electron transfer.	[60,65]
	The anode potential could regulate the accumulation of *c*-type cytochromes at the bacteria–electrode interface, leading to a sudden increase in current followed by a gradual decrease over time.	[86]
	As the voltage increased, there was a significant upregulation in the expression of key EET proteins and elongation factor involved in protein synthesis	[66]
	Under the more positive potential, the reduction of coulomb efficiency is attributed to the direct damaged electron transfer proteins.	[68]
Microbial community	Elevated anode potentials can enhance the abundance of microbial communities in the anode biofilm	[87]
	Electrode potential can screen microorganisms with electrochemical activity.	[92]

**Table 3 microorganisms-13-00631-t003:** Summary of anode potential effects on microbial physiology.

Physiological Aspect	Key Findings	Ref.
Mobility	Applied electric potentials decrease the swimming speed of bacteria but enhance the directionality of movement.	[66,94]
	Electrokinesis enhances the transient swimming speed of bacteria due to near-electrode electron flow.	[67]
Adhesion and Growth	Lower potentials favor microorganisms dominated by DET, while higher anode potentials favor microorganisms by IET or non-electroactive microorganisms.	[58,60].
	Higher anode potentials reduce cell length and increase doubling times.	[97]
	Applied electric fields induce bacteria to produce conductive appendages.	[43]
Metabolism	High potentials activate NADH-dependent pathways, induce branching in TCA cycle, and upregulate the expression of metabolism-related genes.	[70,72,73,98]
	TCA cycle was deactivated at excessive high potential.	[71]
Biofilm	Applied anode potentials promote accumulation of biomass.	[68,76]
	The applied mode of the anode potential and the magnitude of the external resistance also influence the flatness and compactness of biofilms.	[104,105]
	Anode potential affects the proportion of conductive substances and non-conductive polysaccharides in bacterial extracellular polymers.	[61,69,77]

## Data Availability

This review article contains no newly created data.

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
