# Peer review of "The Role of Anode Potential in Electromicrobiology"

_microorganisms, 2025, doi:10.3390/microorganisms13030631_

Round 1

Reviewer 1 Report

Comments and Suggestions for Authors

Comment on the paper: “The Role of Anode Potential in Electromicrobiology” by Yanran Li et al.

General comment:

The manuscript deals with an overview of electroactive microorganisms and their electron transfer mechanisms, elucidates the impact of anode potential on the bioelectricity behaviour and physiology of electroactive microorganisms.

The review is suitable for publication in this journal; however, some points should be addressed before publication.

Some minor language mistakes are present that should be corrected.

  1. Effect of Anode Potential on Electricity Generation Behavior

Please summarize the main findings in tables in order to be more systematic and support the readability of the review.

Please improve the description of this section with a critical point of view.

Please avoid being excessively qualitative, but include quantitative data. For example, please highlight the scale of the application and the performance.

  1. Effect of Anode Potential on Physiology

Please summarize the main findings in tables in order to be more systematic and support the readability of the review.

Please improve the description of this section with a critical point of view.

Author Response

Comment: The manuscript deals with an overview of electroactive microorganisms and their electron transfer mechanisms, elucidates the impact of anode potential on the bioelectricity behaviour and physiology of electroactive microorganisms.

The review is suitable for publication in this journal; however, some points should be addressed before publication.

Response: We thank Reviewer 1 for his/her positive comments and recommendation of our work for publication in Microorganisms after revisions. In the revised manuscript, we have modified text and supplemented tables following his/her suggestions. Details are below.

Comment (1-1): Some minor language mistakes are present that should be corrected.

Response (1-1): Thank you for your careful review and feedback. We have carefully revised the manuscript to correct the minor language mistakes identified. We appreciate your attention to detail and believe that these revisions have improved the clarity and readability of the manuscript.

Page 1, line 32, replaced “chosing” to “choosing”.

Page 2, line 70, replaced “participated” to “participate”.

Page 2, line 78, replaced “faradic” to “faradaic”.

Page 6, line 207, replaced “anode-respiratory” to “anode-respiring”.

Page 6, line 217, replaced “jmax” to “the jmax”.

Page 10, line 331, replaced “c-type” to “c-type”.

Comment (1-2): 4.Effect of Anode Potential on Electricity Generation Behavior

Please summarize the main findings in tables in order to be more systematic and support the readability of the review.

Please improve the description of this section with a critical point of view.

Please avoid being excessively qualitative, but include quantitative data. For example, please highlight the scale of the application and the performance.

Response (1-2):

We sincerely thank the reviewers for valuable comments on this article. Following the reviewer’s suggestion, we have summarized the main findings in Table 2, and improved the description by adding a new sub-section, “4.4 Summary of Anode Potential Effects on Electricity Generation Behavior”. We have also incorporated quantitative data as suggested.

Page 12, paragraph 3, line 408-430

4.4. Summary of Anode Potential Effects on Electricity Generation Behavior

The anode potential serves as a master variable in microbial electrochemical systems, intricately shaping electron transfer efficiency, and community dynamics. A summary of key findings is presented in Table 2. The majority of research indicated that a more positive anodic potential can increase current density and reduce start-up time. However, the optimal anode potential varies depending on the specific microbial strains, electrode materials, and system configurations. For instance, Geobacter species tend to perform better at lower potentials, while Shewanella species show higher activity at moderate potentials. But mechanistically, the influence of anode potential on the electron transfer paths and the dynamic change of microcolonies is similar in these two cases. From the perspective of electron transfer path, as the anode potential rises, electroactive microorganisms undergo several adaptive changes to optimize energy conversion and metabolic processes, thus boosting extracellular electron transfer (EET) efficiency. These changes entail an increase in the abundance of EET proteins and factors involved in protein synthesis, the enrichment of c-type cytochromes on the electrode surface, and the generation of new configurations of c-type cytochromes or the selection of novel EET paths. However, when the anode potential is too high, the structural integrity of c-type cytochromes may be impaired, and the expression of genes related to EET may be further modified, decreasing their EET capacity. Another aspect from which the effects of anode potential on electricity generation behavior can be reflected is the microbial community. Elevated anode potentials could select microorganisms with electrochemical activity and increase the abundance of microbial communities in the anode biofilm.

Table 2. Summary of anode potential effects on electricity generation behavior

Electricity generation aspect

Key findings

Ref.

Start-Up Time and Electric Current/Power Density

A more positive anodic potential can increase current density and reduce start-up time.

[60, 66]

Electron transfer pathways

Electrode potential induces structural transformations of cytochrome to form specific electron transfer pathways, thereby enhancing electron transfer.

[60, 65]

The anode potential could regulate the accumulation of c-type cytochromes at the bacteria-electrode interface, leading to a sudden increase in current followed by a gradual decrease over time.

[86]

As the voltage increased, there was a significant upregulation in the expression of key EET proteins and elongation factor involved in protein synthesis

[66]

Under the more positive potential, the reduction of coulomb efficiency is attributed to the direct damaged electron transfer proteins.

[68]

Microbial community

Elevated anode potentials can enhance the abundance of microbial communities in the anode biofilm

[87]

Electrode potential can screen microorganisms with electrochemical activity.

[92]

Page 7, line 245-246

“So, they achieved a 41% reduction in start-up time and 614.3% increase in current output at +0.2 V in a two-chamber MFC.”

Page 11, line 379-390

“For example, Zhao et al. utilized glucose as a model substrate to investigate the impact of different anode potentials (-0.15 V, 0 V and 0.2 V) on current generation, electron transfer pathways and the abundance of electricity-producing microorganisms within a two-chamber H-type MES reactors with a working volume of 250 mL [87]. Their findings revealed that elevated anode potentials (0 V and 0.2 V) can modulate the degradation pathways of complex substrates within MESs, thereby enhancing current generation from 0.07 mA/cm² to 0.12 mA/cm² and columbic efficiency from 11% to 22% during microbial domestication. Moreover, they observed a direct correlation between the magnitude of current generation and the abundance of microbial communities in the anode biofilm. The abundance of the microbial community in the anode biofilm in-creased by 29% when the anode potential was raised from -0.15 V to 0 V, and by 39% when increased to 0.2 V [87].”

Comment (1-3): 5.Effect of Anode Potential on Physiology

Please summarize the main findings in tables in order to be more systematic and support the readability of the review.

Please improve the description of this section with a critical point of view.

Response (1-3):

We sincerely thank the reviewers for valuable comments on this article. According to your suggestion, we summarized the main findings in Table 3 of new section “5.5. Summary of Anode Potential Effects on Microbial Physiology”, and improved the description to in critical point of view.

Page 17, line 634-646

“5.5. Summary of Anode Potential Effects on Microbial Physiology

Electroactive microorganisms exhibit dynamic physiological adaptations under varying anode potentials, influencing their motility, adhesion, metabolism, and biofilm formation. Reported studies showed that applied anode potentials can modulate motility (promoting or impeding swimming on specific conditions), adhesion (selecting species, reducing cell size/growth rate, and inducing conductive appendages under high potentials), and metabolism (enhancing activity and pathway selection, though inhibited at extremes). Biofilm biomass and structure align with potential levels, bal-ancing conductive and non-conductive extracellular polymers. These responses are critical for optimizing microbial electrochemical systems (MESs). A summary of key findings is presented in Table 3.

Table 3. Summary of anode potential effects on microbial physiology.

Physiological aspect

Key findings

Ref.

Mobility

Applied electric potentials decrease the swimming speed of bacteria but enhance the directionality of movement.

[66, 94]

Electrokinesis enhances the transient swimming speed of bacteria due to near-electrode electron flow.

[67]

Adhesion

& Growth

Lower potentials favor microorganisms dominated by DET, while higher anode potentials favor microorganisms by IET or non-electroactive microorganisms.

[58, 60].

Higher anode potentials reduce cell length and increase doubling times.

[97]

Applied electric fields induce bacteria to produce conductive appendages

[43]

Metabolism

High potentials activate NADH-dependent pathways, induce branching in TCA cycle, and upregulate the expression of metabolism-related genes.

[70, 72, 73, 98]

TCA cycle was deactivated at excessive high potential

[71]

Biofilm

Applied anode potentials promote accumulation of biomass biomass

[68, 76]

The applied mode of the anode potential and the magnitude of the external resistance also influence the flatness and compactness of biofilms

[104, 105]

Anode potential affects the proportion of conductive substances and non-conductive polysaccharides in bacterial extracellular polymers.

[61, 69, 77]

Page 13, line 450-462

“While anode potential serves as a critical regulatory factor, modulating the motility and chemotaxis of electroactive microorganisms, its effects are context-dependent and often contradictory across studies, raising questions about underlying mechanisms and ecological implications. For example, Grobbler et al. discovered that S. oneidensis cultured at -0.19 V exhibited increased abundance of proteins associated with chemotaxis and motility compared to cultures at higher anode potentials of +0.71 V and +0.21 V, suggesting energy-limited conditions drive exploratory behavior [66]. Conversely, Harris et al. reported "electrokinesis" – a transient surge in S. oneidensis swimming speed near electrodes at potentials from -0.6 V to +0.6 V - which paradoxically coexists with reduced chemotaxis protein abundance at higher potentials [67]. This discrepancy implies that motility regulation under anode potential may involve redox-sensitive signaling pathways in-dependent of protein synthesis, yet such mechanisms remain unverified.”

Page 14, line 468-471

“These differences likely stem from variations in surface charge, membrane composition, or flagellar architecture, but no study has systematically correlated bacterial surface properties with motility responses to anode potential.”

Page 14, line 474-485

“However, most studies rely on endpoint protein assays or bulk motility measurements, failing to resolve real-time behavioral dynamics at the electrode-biofilm interface. For example, electrokinesis was only observed in cells closest to electrodes [67], suggesting spatial heterogeneity in motility regulation that conventional methods overlook. Advanced tools like in situ electrochemical microscopy or single-cell tracking could bridge this gap but are rarely applied. Additionally, pilus-mediated twitching motility and secretion-directed chemotaxis are theoretically also influenced by the electrical or chemical gradients generated by electrodes [95, 96]. Conductive pili in Geobacter facilitate both electron transfer and surface attachment, suggesting a dual role that may create trade-offs between motility and electroactivity under varying potentials. Those remain largely unexplored but significantly affects the bacterial ability to adapt to their surroundings and form biofilms.”

Page 14, line 496-499

“However, it is important to consider that the relationship between electrode potential and microbial adhesion is not universally applicable and may vary depending on the specific microorganisms and environmental conditions.”

Page 15, line 504-506

“This dichotomy raises concerns about the universality of "optimal" potentials, as re-al-world systems often require balancing electroactivity with community diversity for long-term stability.”

Page 15, line 511-513

“Yet, such studies rarely address whether these effects stem from specific ion flux alter-ations or broader metabolic reprogramming.”

Page 15, line 517-528

“While this highlights adaptive conductive appendage remodeling, the energy cost of nanowire production and its trade-offs with other cellular functions (e.g., motility, substrate uptake) remain unquantified. Current investigations into the effects of anodic potential on the adhesion and growth of electroactive microorganisms remain in-adequately systematic, with many studies relying on extrapolated conclusions derived from assumptions. To address these limitations, the integration of multivariate optimization strategies and advanced in situ metabolomics profiling emerges as a promising approach to elucidate the intricate relationship between defined potential gradi-ents and adhesion-growth dynamics. Such methodologies could provide mechanistic insights into energy reallocation patterns and biofilm-electrode interfacial adaptations under electrochemically modulated conditions, thereby bridging critical knowledge gaps in the field of electrochemical-microbial interactions.”

Page 15, line 543-545

“This adaptive flexibility is crucial for their survival in diverse electrochemical environments and holds significant implications for the optimization of microbial electro-chemical systems.”

Page 17, line 605-608

“These findings suggest that the physical characteristics of biofilms can be manipulated through the control of anode potential and external resistance, but the long-term stability and functionality of these biofilms under varying conditions need further investigation.”

Page 17, line 619-621

“These contrasting findings highlight the species-specific responses to anode potential, emphasizing the need for a tailored approach to biofilm optimization based on the microbial species involved.”

Page 17, line 631-633

“However, the mechanisms underlying these regulatory processes are not fully under-stood and require further research to develop a comprehensive understanding of bio-films in MESs.”

Reviewer 2 Report

Comments and Suggestions for Authors

The manuscript deals with a scientific topic of high interest and a substantially literature production. Authors have been satisfactorily developed their study, but there is still room for organizational and argumentation improvements prior to publication and, to this end, the proposed review comments can be considered.

  1. In the Abstract section selected numerical data or quantitative information that are related to the conducted review study: years of coverage, databases utilized to collect the studying material, number of documents under certain words used, or any other quantitative information that is related to the key-entities of electromicrobiology interest that arementioned in Figure 1. Classification of electroactive microorganisms and their species.

  1. The critical aspects of the review study have been fully developed and satisfactorily conveyed in sections 4 and 5, together with the fairly well developed Table 1. In this context, and taking into consideration authors’ research objective that “This review provides a summary on the progress in understanding the role of anode potential in electromicrobiology, with a focus on how anode potential impacts electricity generation and physiological behaviors of electroactive microorganisms. The diverse species of electroactive microorganisms and their varied metabolic and EET pathways determine the complexity of the anode potential's influence.”, at the end of section 5 authors are recommended to a new and autonomous 6. Discussion section in which the main findings and the key-aspects of the derived outcomes to be integrated provided and coherently addressed in it. Besides, the research constraints, the challenging issues and the future research paths can be termed in the form of Figure showing a flow chart, diagram, word-cloud, in which the key-findings to be graphically shown. The critical point here is authors to make their own proposition, based on their review study, regarding the statement “synthetic biology and systems biology approaches will be used to modify and optimize microbial strains to enhance their adaptability to potential changes and efficiency of electricity production. For practical purposes, future studies focusing on integrating potential modulation”. Actually this Discussion section can enhance and expand the novelty of authors’ review study, synthesizing and composing the separate studies under a coherent and comprehensive entity/unity.

Author Response

Comment: The manuscript deals with a scientific topic of high interest and a substantially literature production. Authors have been satisfactorily developed their study, but there is still room for organizational and argumentation improvements prior to publication and, to this end, the proposed review comments can be considered.

Response: We thank Reviewer 1 for his/her positive comments and recommendation of our work for publication in Microorganisms after revisions. In the revised manuscript, we have modified text and supplemented figures following his/her suggestions. Details are below.

Comment (2-1): In the Abstract section selected numerical data or quantitative information that are related to the conducted review study: years of coverage, databases utilized to collect the studying material, number of documents under certain words used, or any other quantitative information that is related to the key-entities of electromicrobiology interest that arementioned in Figure 1. Classification of electroactive microorganisms and their species.

Response (2-1): Thank you for your valuable comments and suggestions. We have carefully considered your feedback and added specific details regarding the scope and methodology of our review study to Abstract

Page 1, Abstract, line 27-28:

“In this review, we have summarize approximately 100 relevant studies, sourced from PubMed and Web of Science over the past two decades.”

Comment (2-2): The critical aspects of the review study have been fully developed and satisfactorily conveyed in sections 4 and 5, together with the fairly well developed Table 1. In this context, and taking into consideration authors’ research objective that “This review provides a summary on the progress in understanding the role of anode potential in electromicrobiology, with a focus on how anode potential impacts electricity generation and physiological behaviors of electroactive microorganisms. The diverse species of electroactive microorganisms and their varied metabolic and EET pathways determine the complexity of the anode potential's influence.” at the end of section 5 authors are recommended to a new and autonomous 6. Discussion section in which the main findings and the key-aspects of the derived outcomes to be integrated provided and coherently addressed in it. Besides, the research constraints, the challenging issues and the future research paths can be termed in the form of Figure showing a flow chart, diagram, word-cloud, in which the key-findings to be graphically shown. The critical point here is authors to make their own proposition, based on their review study, regarding the statement “synthetic biology and systems biology approaches will be used to modify and optimize microbial strains to enhance their adaptability to potential changes and efficiency of electricity production. For practical purposes, future studies focusing on integrating potential modulation”. Actually this Discussion section can enhance and expand the novelty of authors’ review study, synthesizing and composing the separate studies under a coherent and comprehensive entity/unity.

Response (2-2): We sincerely appreciate the reviewer’s insightful suggestion to strengthen the novelty and coherence of the Discussion. Based on reviewer’s suggestions, we have expanded the section 6 by adding a conceptual framework titled “Smart Voltage-Responsive Communities” as Figure 6 in the revised manuscript which integrates electrode engineering, synthetic biology, and systems biology to address current challenges, and visually synthesizes the separate studies into a unified strategy. The associated discussions were also added.

Page 19, line 681-698:

“Here, we propose a “Smart Voltage-Responsive Communities” framework (Figure 6), where electrode materials are innovated to achieve corrosion-resistant and highly conductive activity, synthetic biology tools enable the design of voltage-responsive microbial strains, while systems biology deciphers metabolic-electrochemical coupling under varying potentials. For instance, electrode materials integrate bioresponsive interfaces (e.g., graphene-cytochrome hybrids) to synchronize potential modulation with microbial activity. Embedding voltage-sensitive promoters into electroactive microorganisms could dynamically regulate cytochrome expression, achieving adaptive electron transfer. Additionally, establishing a voltage gradient forms spatially stratified communities (such as high-potential zones and low-potential zones). Concurrently, machine learning models trained on multi-omics datasets may predict optimal potential windows for specific microbial communities. Ultimately, this framework aims to bridge lab-scale discoveries and industrial applications, driving the development of next-generation MESs that are both efficient and scalable.”

Figure 6. Proposed framework for advancing microbial electrochemical systems (MESs) through synthetic biology, systems biology, and potential modulation.

Round 2

Reviewer 1 Report

Comments and Suggestions for Authors

The authors revised the manuscript according to the comments/changes suggested. The paper is suitable to be published in this journal in its current form.